# External Load of Flamenco Zap-3 Footwork Test: Use of PlayerLoad Concept with Triaxial Accelerometry

**DOI:** 10.3390/s22134847

**Published:** 2022-06-27

**Authors:** Ningyi Zhang, Sebastián Gómez-Lozano, Ross Armstrong, Hui Liu, Alfonso Vargas-Macías

**Affiliations:** 1Performing Arts Research Group, Faculty of Sport, San Antonio Catholic University, 30107 Murcia, Spain; nzhang@alu.ucam.edu; 2Rehabilitation and Healthy Lives Research Group, Institute of Health, University of Cumbria, Carlisle CA1 2HH, UK; ross.armstrong@cumbria.ac.uk; 3Biomechanics Laboratory, Beijing Sport University, Beijing 100084, China; liuhui@bsu.edu.cn; 4Telethusa Centre for Flamenco Research, 11004 Cádiz, Spain; vargas@flamencoinvestigacion.es

**Keywords:** triaxial accelerometry, dance, flamenco footwork, external load, PlayerLoad

## Abstract

The intense footwork required in flamenco dance may result in pain and injury. This study aimed to quantify the external load of the flamenco Zapateado-3 (Zap-3) footwork via triaxial accelerometry in the form of PlayerLoad (PL), comparing the difference in external loads at the fifth lumbar vertebra (L5), the seventh cervical vertebra (C7) and the dominant ankle (DA), and to explore whether the speed, position, axis and proficiency level of the flamenco dancer affected the external load. Twelve flamenco dancers, divided into professional and amateur groups, completed a 15-s Zap-3 footwork routine at different speeds. Triaxial accelerometry sensors were positioned at the DA, L5 and C7 and were utilized to calculate the total PlayerLoad (PLTOTAL), uniaxial PlayerLoad (PLUNI) and uniaxial contributions (PL%). For both PLTOTAL and PLUNI, this study identified significant effects of speed and position (*p* < 0.001), as well as the interaction between speed and position (*p* ≤ 0.001), and at the DA, values were significantly higher (*p* < 0.001) than those at C7 and L5. Significant single axis and group effects (*p* < 0.001) and effects of the interactions between the position and a single axis and the group and speed (*p* ≤ 0.001) were also identified for PLUNI. Medial-lateral PL% represented a larger contribution compared with anterior-posterior PL% and vertical PL% (*p* < 0.001). A significant interaction effect of position and PL% (*p* < 0.001) also existed. In conclusion, the Zap-3 footwork produced a significant external load at different positions, and it was affected by speed, axis and the proficiency level of the flamenco dancer. Although the ankle bears the most external load when dancing the flamenco, some external load caused by significant vibrations is also borne by the lumbar and cervical vertebrae.

## 1. Introduction

Flamenco dance is performed with strong emotional expression, and its footwork techniques, which include striking the floor with a loud and rhythmic sound, make it different from other dance genres [1,2]. The physical effort demanded in performing flamenco is similar to that of elite sports [3,4,5,6]. Therefore, practicing the required footwork in flamenco may cause chronic repetitive pain and injuries of the feet, knees and spine, mainly at the lumbar and cervical levels [4,7,8,9,10,11,12]. Previous studies regarding flamenco’s technical movements have focused on analyzing them from the perspective of electromyography, kinematics and epidemiology [9,13,14,15]. However, questionnaires and labor-intensive methods were often used in these studies. Normally, labor-intensive methods test one subject and cannot provide real-time live feedback [16]. These shortcomings, along with the vast amounts of data downloaded and analyzed in a single study [17], have raised questions about the ability of these technologies to influence everyday dance practices.

Triaxial accelerometers are motion sensors used to detect movement in three planes of movement (medial-lateral, anterior-posterior and vertical) and therefore to provide data regarding the magnitudes and frequency of movement. The concept of PlayerLoad (PL) enables the measurement of a vector modified algorithm proposed by the technological company Catapult Sports, utilizing a micro-electrical mechanical system. It is expressed as the square root of the sum of the squared instantaneous rates of change in acceleration for each of the three vectors (medial-lateral, anterior-posterior and vertical) divided by 100. This technology requires highly responsive motion sensors to record movement along these vectors. The micro-electrical mechanical system device contains a triaxial piezoelectric linear accelerometer that samples at a frequency of 150 Hz, therefore providing the opportunity to quantify movement performance. Due to its low user dependence, PL has been used in various physical activity tests to describe the external load [18,19,20,21] and is associated with sports training and competition for a broad range of athletes. Within dance research, PL has also been used extensively to quantify the external load or mechanical load and the relationship with dance injuries [22,23,24]. Nagy [25] investigated within-day and between-day loading responses to ballet choreography and reported that PL is sufficiently sensitive for use with a progressive routine and that accelerometers are effective for athlete monitoring and injury screening protocols, supporting previous work indicating that triaxial PL was sensitive enough to detect the increased loading associated with increases in exercise intensity when quantifying PL, PLUNI and PL% [17].

In summary, future research regarding the external load during the footwork routines of flamenco dance is required to provide medical practitioners, coaches and dancers with a theoretical basis for the effective management of training programs, to reduce injury risk in accordance with the proficiency levels of dancers and to provide a feasible method for assessing flamenco footwork techniques. Furthermore, Zapateado-3 (Zap-3) is a topical issue surrounding flamenco footwork. It is widely used to analyze flamenco technique and associated movements [4,13,20]. Therefore, this study aimed to quantify the external load during performance of the Zap-3 footwork technique via triaxial accelerometry in the form of PL values, comparing the difference in external load at a lumbar vertebra, a cervical vertebra and the dominant ankle, and to explore whether speed, position, axis and the proficiency level of the flamenco dancer affected the external load.

## 2. Materials and Methods

### 2.1. Participants

Twelve flamenco dancers volunteered for this study. They were recruited by via posters promoting the study in three dance institutions that included flamenco dance training and performance. The participants’ demographics are reported in Table 1. The procedures, risks and benefits of the test were explained to the participants in advance. Participants provided informed consent prior to testing. Ethical approval was granted by the Sports Science Experiment Ethics Committee of Beijing Sport University (2022037H), and the study was completed in accordance with the Declaration of Helsinki.

The participants consisted of a professional group (group P, 6 participants) and an amateur group (group A, 6 participants). The inclusion criteria for group P were that participants had to be professional flamenco dancers who received paid work for teaching, rehearsing or performing in the flamenco dance field and who primarily considered themselves to be professional flamenco dancers with a minimum of 3 years’ experience. For group A, participants had to be amateur flamenco dancers who engaged in dance for recreational purposes only and attended flamenco dance training for at least 3 h per week. All participants were over 18 years of age and had had no musculoskeletal injuries in the 6 months preceding the test.

### 2.2. Procedures

All participants were informed about the experimental methods and procedures, and the flamenco techniques were demonstrated by a teacher with 12 years’ experience as a qualified flamenco dance teacher. Accelerometer application was performed by a laboratory technician with 5 years’ experience and training in the use of accelerometers. The process order was fixed for each participant. Each participant was required to perform the Zap-3 footwork at 160 bpm (beats per minute), 180 bpm and the fastest (as fast as they could) speed level, in sequence, on the same flamenco dance folding portable floor (measuring 0.92 × 1 m^2^, made of wood) in a dance studio. Each speed level was presented 3 times for a duration of 15 s. At 160 bpm and 180 bpm, participants had to strike the floor twice on each beat, utilizing earphones that played a metronome. At the fastest level (F), the sound had to be rhythmic, and the frequency at each speed level is reported in Table 2. Group P demonstrated a significantly higher speed than Group A (*p* < 0.05). Participants were allowed to practice 3 times before the data were recorded, to facilitate adaptation to the next speed level, and they were allowed a 3-min rest between recordings to reduce fatigue. Participants were asked to perform wearing flamenco shoes that were similar to those they wore during performances, rehearsals or daily training. During the entire footwork movement, participants were required to maintain an akimbo posture with their hands (Figure 1), to keep their upper limbs and trunk stable and to perform smooth and coherent movements. Figure 2 provides a flowchart of the procedure for the Zap-3 footwork test.

### 2.3. Flamenco Zap-3 Footwork

Participants completed the sequence of flamenco Zap-3 footwork, which is a sequence of 6 steps completed bilaterally. When one sequence is completed, it is repeated with the other foot, and this repetition continues with alternating feet [4,13]. The 6 steps are:(1)Zapateado de planta (P);(2)Zapateado de Tacón-planta (TP);(3)Zapateado de Tacón (T);(4)Zapateado de Tacón-planta (TP);(5)Zapateado de Punta (PNT);(6)Zapateado de Tacón-planta (TP).

### 2.4. Data Processing

Trigno Avanti™ sensors (Trigno Wireless EMG System, Delsys, Natick, MA, USA), which have a built-in nine-degrees-of-freedom inertial measurement unit and can relay acceleration, rotation and earth magnetic field information, were utilized to record the flamenco Zap-3 footwork’s external load responses, with data sampling at 150 Hz. The sensors were positioned at the 5th lumbar vertebra (L5), the 7th cervical vertebra (C7) and the dominant ankle (DA) [10,17,25,26,27,28,29]. The dominant foot was defined as the foot which participants would most often use to kick a ball [29,30]. The sensors were attached directly to the skin using medical tape and secured using elastic bandage. The locations were determined by palpation, and the ankle location was 1cm proximal to the lateral malleolus.

The uniaxial PlayerLoad (PLUNI) was calculated as the square root of the instantaneous rate of change in acceleration in each of the medial-lateral (PLML), anterior-posterior (PLAP) and vertical (PLV) planes divided by 100. The accumulated total PlayerLoad (PLTOTAL) was defined as the square root of the sum of the squared instantaneous rates of change in acceleration in each of the three planes divided by 100 and was calculated at L5, C7 and the DA. The uniaxial contributions (PL%), defined as the percentage contributions of the PLUNI in the medial-lateral (PLML%), anterior-posterior (PLAP%) and vertical (PLV%) planes, was also quantified by dividing the individual PLUNI value by PLTOTAL and by multiplying that value by 100.

### 2.5. Statistical Analysis

Data were analyzed using SPSS (SPSS IBM Statistics V21.0, IBM, Armonk, New York, NY, USA). Descriptive statistics are reported as means ± standard deviations. The descriptive characteristics of age, height, mass, body mass index (BMI) and years of experience dancing flamenco, together with the frequency of the F speed level, were analyzed between group P and group A using a Mann-Whitney U test, as the dependent variable was not normally distributed. The assumptions of normality were verified using the Shapiro-Wilk test. Differences in each dependent variable during the Zap-3 footwork were quantified using a general linear model (GLM). Bonferroni correction factors were used for a post hoc comparison, to determine where any significant differences occurred. The 95% confidence intervals (CIs) and Cohen’s *d* effect sizes were as follows: small, 0.20–0.49; moderate, 0.50–0.79; large > 0.80 [31,32]. Statistical significance was set at the *p* < 0.05 level.

## 3. Results

### 3.1. Zap-3 Footwork Load Responses—Total PlayerLoad

Figure 3 and Figure 4 report the results for different speed levels for PLTOTAL for the Zap-3 footwork in the professional and amateur groups. A significant main effect was identified for speed (*p* < 0.001), and the value of PLTOTAL increased with speed level. There was no significant main effect for the groups (*p* > 0.05). There was also no significant group x speed interaction (*p* > 0.05).

Significant main effects for unit position (*p* < 0.001) as well as for the speed x position interaction (*p* ≤ 0.001) were identified. PLTOTAL values were higher for the DA (212.98 ± 72.27 au; CI = 201.30–224.66 au) compared with C7 (38.27 ± 14.70 au; CI = 26.59–49.95 au; *p* < 0.001; *d* = 3.35) and L5 (44.15 ± 17.68 au; CI = 32.47–55.83 au; *p* < 0.001; *d* = 3.21), with the differences becoming more pronounced as the speed increased. There was no significant difference between L5 and C7 (*p* > 0.05). There was a significant difference for the DA at different speed levels (*p* < 0.001), but this difference was not identified in the L5 and C7 positions (*p* > 0.05). There was no significant interaction between group and position (*p* > 0.05) or between group, speed and position (*p* > 0.05).

### 3.2. Zap-3 Footwork Load Responses—Uniaxial PlayerLoad

Significant main effects for unit position (*p* < 0.001) and a single axis (*p* < 0.001) were identified (Figure 5), and there was also a position x single axis interaction effect (*p* < 0.001). The PLML values were higher at the DA (113.33 ± 39.20 au; CI = 107.28–119.38 au) compared with C7 (28.94 ± 12.14 au; CI = 22.89–34.99 au; *p* < 0.001; *d =* 2.91) and L5 (18.17 ± 6.80 au; CI = 12.12–24.22 au; *p* < 0.001; *d* = 3.38). A significant difference existed between L5 and C7 (*p* < 0.05; *d* = 1.09). 

The PLV values were higher at the DA (130.49 ± 42.62 au; CI = 124.44−136.54 au) compared with C7 (8.31 ± 3.95 au; CI = 2.26–14.36 au; *p* < 0.001; *d* = 4.04) and L5 (22.57 ± 10.71 au; CI = 16.52–28.62 au; *p* < 0.001; *d* = 3.47). There was a significant difference between L5 and C7 (*p* < 0.01; *d* = 1.77).

The PLAP values were higher at the DA (69.31 ± 28.85 au; CI = 63.26–75.36 au) compared with C7 (15.02 ± 5.72 au; CI = 8.97–21.07 au; *p* < 0.001; *d* = 2.61) and L5 (22.14 ± 8.93 au; CI = 16.36–28.46 au; *p* < 0.001; *d* = 2.21). There was no significant difference between L5 and C7 (*p* > 0.05).

There was a significant different in the DA position for different axes (*p* < 0.001), but this difference was not identified in L5 (*p* > 0.05). At C7, there was a significant difference between PLML and PLV (*p* < 0.001) and between PLML and PLAP (*p* < 0.01) but no difference between PLV and PLAP (*p* > 0.05).

Significant main effects were identified for the speed (*p* < 0.001) and group (*p* < 0.01). There was significant group x speed interaction effect (*p* ≤ 0.001). Post hoc analyses revealed that there was a significant difference between groups at the fastest speed levels (*p* < 0.05), but this difference was not found at the 160 bpm or 180 bpm speed levels (*p* > 0.05). There was a difference in group P at different speeds (*p* < 0.001), as well as in group A (*p* < 0.001). 

A speed x position interaction (*p* < 0.001) was also identified. There was a significant difference in ankle loading at different speed levels (*p* < 0.001), but this difference was not identified in L5 and C7 (*p* > 0.05), and there was a significant difference between DA and L5 (*p* < 0.001) and between DA and C7 (*p* < 0.001) but no significant difference between L5 and C7 (*p* > 0.05) at any speed level.

### 3.3. Zap-3 Footwork Load Responses—Uniaxial Contributions

There was no significant main effect for group (*p* = 0.841) or speed (*p* = 0.739) for any axis. Although the statistical significance value indicated a significant difference (*p* < 0.001), the Cohen’s *d* effect sizes between positions were smaller than 0.2, showing no significant main effect differences for the position.

There was a significant main effect identified for PL% (*p* < 0.001), with PLML% (56.75 ± 14.59%; CI = 55.70–57.80) representing a significantly larger contribution compared with PLV% (44.25 ± 17.92%; CI = 43.20–45.30; *p* < 0.001; *d* = 0.76) and PLAP% (41.97 ± 9.47; CI = 39.90–42.00; *p* < 0.001; *d* = 1.20). The PLAP% values were lower than the PLV values (*p* < 0.001; *d* = 0.16). Although the difference between PLAP% and PLV% was statistically significant (*p* < 0.001), the Cohen’s *d* effect sizes (*d* = 0.16) were smaller than 0.2, which demonstrates that there was no significant difference.

Post hoc analysis of a significant unit position x uniaxial interaction contribution (*p* < 0.001) demonstrated that the PLML% at C7 (74.87 ± 5.28%; CI = 73.00–76.70) was significantly greater than that at both L5 (42.26 ± 6.51%; CI = 40.40–44.10; *p* < 0.001; *d* = 5.50) and DA (53.13 ± 3.59%; CI = 51.30–55.00; *p* < 0.001; *d* = 4.82). Moreover, there was a significant difference between DA and L5 (*p* < 0 001; *d* = 2.07) (Figure 6).

The PLV% values at the DA (61.63 ± 3.66%; CI = 59.80–63.50) were significantly higher than at both C7 (21.22 ± 3.26%; CI = 19.40–23.10; *p* < 0.001; *d* = 11.66) and L5 (49.92 ± 8.27%; CI = 48.10–51.80, *p* < 0.001; *d* = 1.83), and those at L5 were significantly higher than those at C7 (*p* < 0 001; *d* = 4.57). The PLAP% values at L5 (51.13 ± 5.40%; CI = 49.30–53.00) were significantly higher than at both DA (31.96 ± 3.60%; CI = 30.10–33.80; *p* < 0 001; *d* = 4.18) and C7 (39.82 ± 6.42; CI = 38.00–41.70; *p* < 0.001; *d* = 1.91), and there was a difference between DA and C7 (*p* < 0.001; *d* = 1.51) (Figure 6). There was no interaction among other effects in uniaxial contributions.

## 4. Discussion

This study aimed to quantify the external load in flamenco Zap-3 footwork via triaxial accelerometry in the form of PlayerLoad (PL), comparing the differences in external load at a lumbar vertebra, a cervical vertebra and the dominant ankle, with consideration of speed, position, axis and the proficiency level of the flamenco dancer. The triaxial PlayerLoad is sensitive enough to detect a load change associated with the intensity of the dance movement [22,33,34] and it is hypothesized that an increase in cumulative accelerometer load may cause injury [17,35]. Consequently, the results of this study could have implications for potential injury risk, from the perspective of variables such as position, speed and the proficiency level of flamenco dancers.

An important finding of this study is that PLTOTAL, PLUNI and PL% were all higher at the DA compared with C7 and L5, suggesting that in flamenco, when practicing footwork techniques, the ankle is subject to higher loading, which consequently might have implications for injury risk. Furthermore, the PLML was higher at the DA, which is potentially due to the normal foot imbalance when practicing the Zap-3 footwork. The PLV was higher at the DA, which may be due to the body positions required while practicing the Zap-3 footwork, which requires more movement in the vertical plane. The PLAP was higher at the DA, which may be due to the flamenco Zapateado technique being performed with knee flexion extensions. Additionally, our study found a significant difference for the DA for different axes, potentially because, during the Zapateado, the foot performs movements on all three axes [11]. Previous studies that used triaxial accelerometry in other dance genres have reported that lower limbs bear higher loads on the three axes than C7, specifically in both dance aerobic fitness tests and ballet choreography [17,25]. Flamenco dance injury studies have reported that the frequency of injury or pain in the foot is higher than in the spine or in other body locations and have highlighted the fact that that the amount of time spent practicing footwork may affect the rate of injury incidence [2,10]. Foot disorders such as metatarsal pain and hyperkeratosis in the forefoot are not an uncommon phenomenon for female flamenco dancers and may be caused by chronic repetitive trauma suffered during footwork practice [8]. Therefore, the higher loads observed at the DA in this study may potentially increase injury risk.

Although the PLTOTAL and PL% values for C7 were lower than for the DA in flamenco Zap-3, they were comparatively higher than in other dance tests. In ballet, a specific choreographed routine test with five stages was performed by 10 participants [25], and the routine’s fifth stage had the highest PL. The highest uniaxial PL and PL% values were lower than our results for the Zap-3 test, and we found that the PLTOTAL and PL% values at C7 were lower than at the DA and L5 for the flamenco Zap-3 footwork test with the fastest speed level in both the professional and amateur groups. Although the speed (beats per minute) in these two tests was slower, they were performed for 4 min; in contrast, the Zap-3 lasted for just 15 s. Therefore, we can deduce that potentially the cervical vertebrae can be loaded to a greater extent during flamenco and could therefore potentially be injured by practicing flamenco footwork. Other flamenco studies have reported a high frequency of injury or pain in the cervical spine [2,10].

For the external load in the lumbar vertebrae, this study found that the PLTOTAL, PLV, PLV% and PLAP values were lower than at the DA but greater than at C7. The PLML and PLML% contribution for L5 was lower than for both C7 and DA, which indicates that increased loading and the potential for injury risk on the ML axis may potentially be less, if load is directly related to injury occurrence. However, consideration of the different mechanisms of injury that can occur at the spine is required, as loading directions might potentially influence injury development. For example, the L4/5 level is the most common location for lumbar spine injury; however, the influence of the direction of force, e.g., anterior-posterior, vertical or medial-lateral is not known, and therefore this could be an important consideration for future research. Previous injury studies of flamenco dancers stated that the lumbar spine is associated with a high risk of injury [10] and that the highest prevalence of spinal injury in flamenco dancers occurs in the lumbar spine [2]. To the best of our knowledge, limited research exists using accelerometer placement at L5 to determine injury risk in dance. However, L5 has been proven to have sufficient sensitivity to detect the external load in this region [26,28,29].

Zap-3 is a symbolic flamenco dance training footwork technique that does not require much movement of the upper limbs or trunk. The displacements of the center of gravity are minuscule: 0.136 m in anteroposterior movement, 0.105 m in lateral and 0.018 m in vertical displacement [36]. In our study, participants were asked to keep their hands in an akimbo posture and to keep their upper limbs and trunk stable; however, we found that significant external loads were recorded at the cervical and lumbar positions. It should be noted that this Zap-3 movement model, which does not engage the upper body, could be the fundamental reason why the spinal structures are forced to absorb vibrations that have not been dissipated during tilts [10]. The biomechanical stomping mechanisms have a similar impact on the musculo-articular kinetic chain. Vibrational waves transmit the impact of the shoe from the joints of the lower body to the spine, which can trigger vertebral pain [37] and overload the spinal muscles [38]. A survey of the injury frequency of 75 flamenco dancers demonstrated that 16% and 22.7% suffered from lumbar and cervical spine injury, respectively. In other forms of percussive dance, back injury was not as prevalent, e.g., in Irish dancers (5% of 159 dancers) [39] and tap dancers (15% of 104 dancers) [12]. Furthermore, cervical injury was not as prevalent in Irish dancers (1% of 159 dancers). Such differences could be due to different performance characteristics and could be explained by jumping during percussions with the foot, which more effectively dissipates the large vibrations.

Regarding differences between professional and amateur flamenco dancers, our study demonstrated that a difference in external load occurs only at the fastest speed level and that the professional group had more external load than the amateurs for uniaxial PL. This difference may be because the professional group had a higher fastest speed and their movement quality was higher. Professional dancers tended to strike the floor firmly to make a louder sound while performing the footwork, producing more ground reaction force [13]. Therefore, professional dancers may have higher injury risk when practicing their footwork. Considering the dancers’ experience, M. Elizabeth Pedersen reported that the number of injuries sustained by professional flamenco dancers was greater than the number sustained by student flamenco dancers [2]. Similar findings have been reported for other dance styles and sports [40,41,42]. Eileen M. Wanke’s group reported a greater asymmetric load in the highest national league group of Latin dancers than in the regional or lower groups, and they were injured more often in their right hands and shoulders [43].

Regarding the difference between the speed levels, our study identified a main effect of speed (*p* < 0.001) in the case of PLTOTAL or uniaxial PL, and PLTOTAL increased with speed level. Additionally, a speed x position interaction (*p* ≤ 0.001) between PLTOTAL and PL was also found. Post hoc analyses revealed that the differences became more pronounced as the speed increased. Furthermore, a group x speed interaction for uniaxial PL was also proven, indicating a significant difference at the fastest level between professional and amateur dancers (*p* < 0.05). This evidence, combined with the differences existing in group, position and axis effect suggests that the accelerometry technique demonstrated sufficient sensitivity during the Zap-3 flamenco dance footwork test.

Accelerometry has been used infrequently in flamenco footwork tests. Literature searches identify only one article, which explored musculoskeletal demands on flamenco dancers more than thirty years ago [9]. In this study, ten dancers performed dance steps with accelerometry sensors located at their tibial tuberosity and the anterior superior iliac spine. Data from these accelerometry sensors were recorded as peak frequencies and amplitudes. The current study involved the concept of PlayerLoad, which is a more modern concept allowing for greater reliability and standardization. Additionally, the use of triaxial accelerometry allowed us to analyze the details of each plane. Furthermore, the triaxial accelerometry sensor used in this study provided a non-invasive way to measure the external loads encountered by a dancer’s body, which helps dancers, teachers and medical staff improve performance quality, adapt training loads and programs and inform on rehabilitation strategies. These are practices that are currently observed in sports with high injury incidence. One of the main contributions of this biomechanical study was the involvement of six professional flamenco dancers, while a previous study analyzed the case of professional flamenco dancers with only one participant, as a case study [13,15,36]. The second contribution was the comparison between the professional and amateur groups, which allowed the assessment of the association between technical progression and biomechanical variables. The last relevant contribution of this research compared to previous studies was that triaxial accelerometers were used for the first time to determine the external load in the form of PlayerLoad in flamenco dance. Study limitations included the relatively small sample size and the fact that only one type of footwork was investigated. Future studies could consider a larger sample and explore the external load at other positions such as the knees and upper limbs and could analyze a complete flamenco dance choreography.

## 5. Conclusions

This study quantified the mechanical demands of the footwork required of flamenco dancers and explored whether speed, position, axis and the proficiency level of the flamenco dancer affected the external load. In conclusion, the Zap-3 footwork produced a significant external load at different positions, which was affected by speed, axis and the proficiency level of the flamenco dancer. Although the ankle bears the highest external load when dancing the flamenco, some external load is also experienced by the lumbar and cervical vertebrae, caused by significant vibrations. This study provides medical practitioners, coaches and dancers with a theoretical basis for the development of an appropriate training program to reduce injury risk and to provide a feasible method for assessing flamenco footwork techniques in future studies.

## Figures and Tables

**Figure 1 sensors-22-04847-f001:**
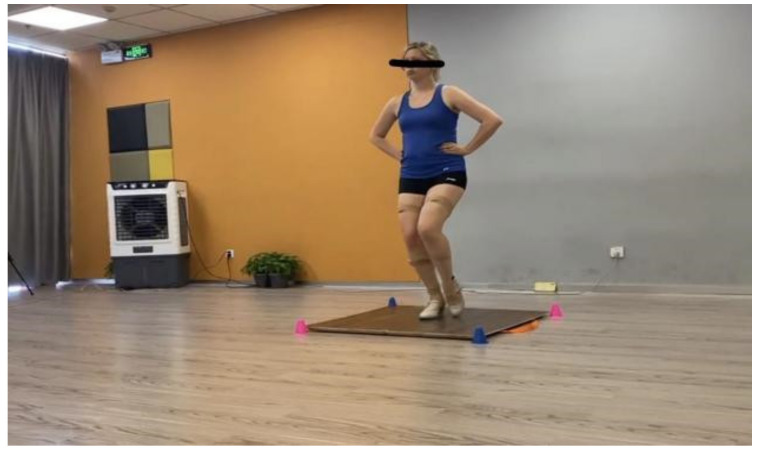
Akimbo posture required for completing the footwork test.

**Figure 2 sensors-22-04847-f002:**
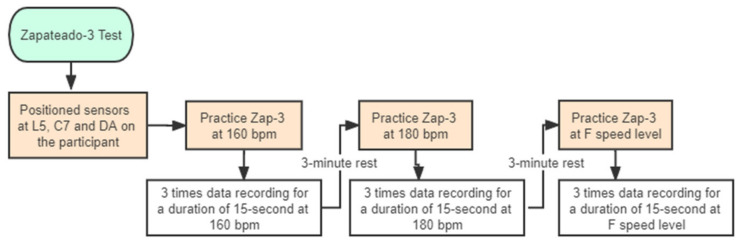
The procedure for the footwork test. L5: the 5th lumbar vertebra; C7: the 7th cervical vertebra; D: the dominant ankle; bpm: beats per minute; F: the fastest speed level.

**Figure 3 sensors-22-04847-f003:**
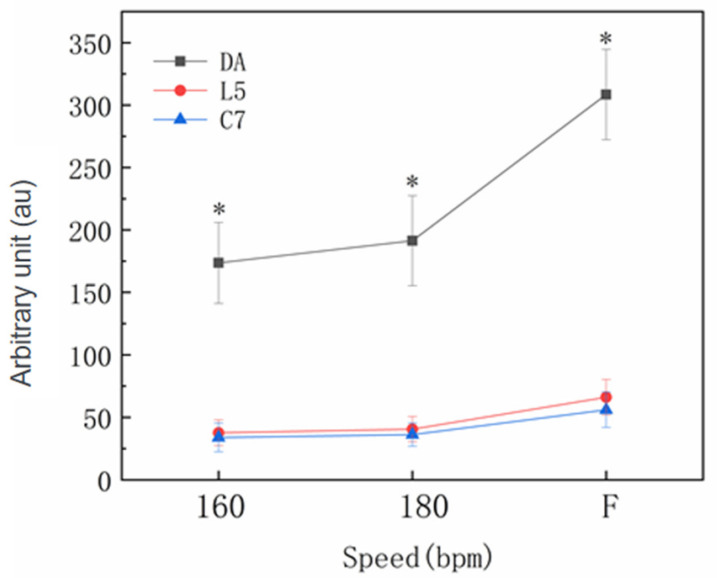
The professional group total PlayerLoad responses to the Zapateado-3 footwork at 160 bpm, 180 bpm and the fastest speed level (F) for the dominant ankle (DA), the fifth lumbar vertebra (L5) and the seventh cervical vertebra (C7). * Denotes a significant main effect for the unit position.

**Figure 4 sensors-22-04847-f004:**
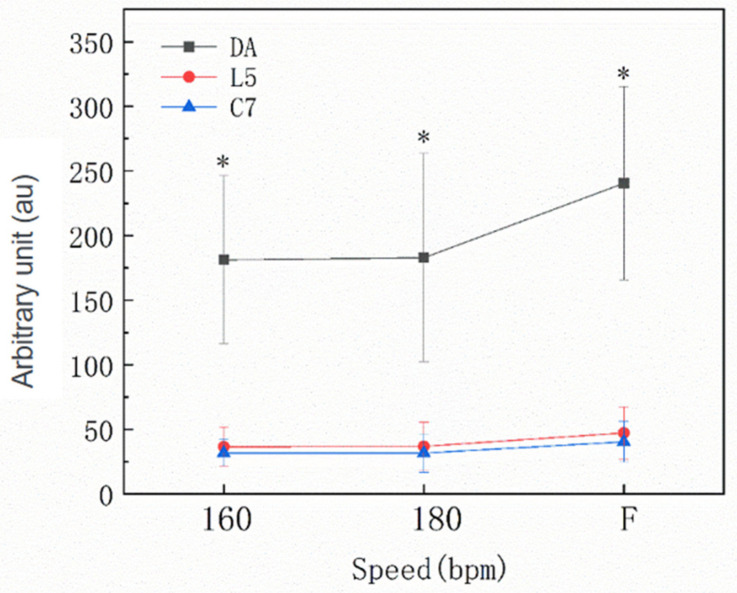
The amateur group total PlayerLoad responses to the Zapateado-3 footwork at 160 bpm, 180 bpm and the fastest speed level (F) for the dominant ankle (DA), the fifth lumbar vertebra (L5) and the seventh cervical vertebra (C7). * Denotes a significant main effect for the unit position.

**Figure 5 sensors-22-04847-f005:**
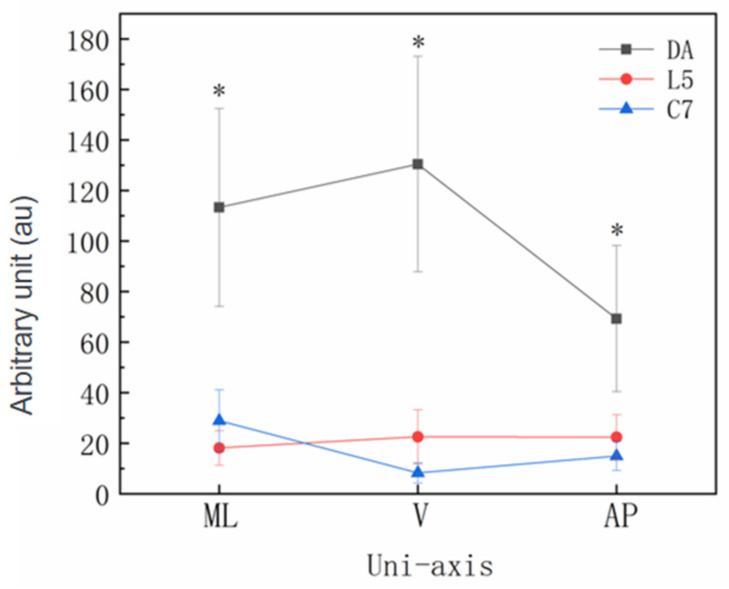
The differences in PlayerLoad responses between the dominant ankle (DA), the fifth lumbar vertebra (L5) and the seventh cervical vertebra (C7) for the Zapateado-3 footwork in three planes: medial-lateral (ML), vertical (V) and anterior-posterior (AP). * Denotes a significant main effect for the unit position.

**Figure 6 sensors-22-04847-f006:**
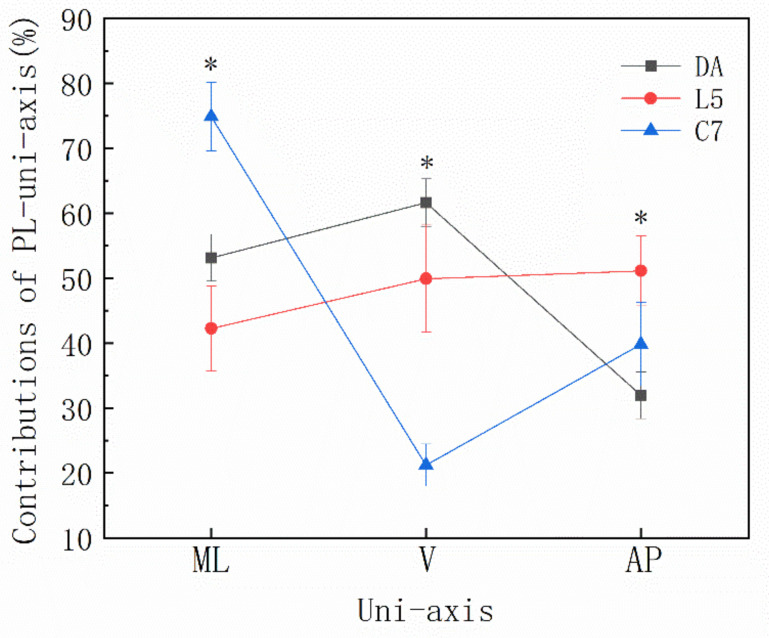
The different uniaxial contributions to the total PlayerLoad responses for the dominant ankle (DA), the fifth lumbar vertebra (L5) and the seventh cervical vertebra (C7) for the Zapateado-3 footwork in three planes: medial-lateral (ML), vertical (V) and anterior-posterior (AP). * Denotes a significant main effect for the position.

**Table 1 sensors-22-04847-t001:** Descriptive characteristics of participants (n = 12).

Characteristics	Group P n = 6	Group A n = 6	*p*
Age (years)	38.83 ± 7.96	34.50 ± 10.67	0.148
Height (m)	1.67 ± 0.10	1.62 ± 0.03	0.681
Mass (kg)	63.33 ± 6.38	56.17 ± 15.99	0.055
BMI (kg/m^2^)	22.79 ± 1.95	21.36 ± 6.00	0.078
Flamenco dance experience (years)	7.67 ± 4.89	1.83 ± 1.17	*** 0.009**

* Denotes a significant difference between groups at the *p* < 0.05 level. kg: kilograms; m: meters; m^2^: square meters.

**Table 2 sensors-22-04847-t002:** Descriptive speed levels for professional and amateur groups (n = 12).

Speed	160 BPM		180 BPM		* F	
Group	Group P	Group A	Group P	Group A	Group P	Group A
Frequency (Hz)	5.33	5.33	6.00	6.00	8.99 ± 0.78	7.08 ± 0.50

* Denotes a significant difference between groups at *p* < 0.05. F: the fastest speed level; Hz: Hertz; BPM: beats per minute.

## Data Availability

The data presented in this study are available on request from the corresponding author.

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
