# Peer review of "External Load of Flamenco Zap-3 Footwork Test: Use of PlayerLoad Concept with Triaxial Accelerometry"

_sensors, 2022, doi:10.3390/s22134847_

Round 1

Reviewer 1 Report

The authors quantified the external load of the flamenco Zapateado-3 (Zap-3) footwork via triaxial accelerometry in the form of PlayerLoad (PL), comparing the difference in external loads at the fifth lumbar vertebrae (L5), the seventh cervical vertebrae (C7) and the dominant ankle (DA). The authors also explored the speed, position, axis and the proficiency level of flamenco dancers affect the external load. The study is interesting, and it fits in the journal. However, the English language has several problems, and it was difficult to read. There are some comments:

1. The references need to be updated with more recent references;

2. The English is the major problem of the manuscript. I suggest using a proofread service;

3. The methods are well-structured, but I suggest to the authors to a flow chart with the description of the implemented method;

4. The results only need a proofread;

5. In the discussion, a table with advantages and disadvantages of the proposed study with the previous studies benefits the manuscript.

Author Response

Thank you for the opportunity to amend the manuscript.

Thanks a lot for your time

Reviewer 2 Report

Comments to the manuscript ID: sensors-1770322 entitled:” External Load of Flamenco Zapateado-3 footwork Use of PlayerLoad Concept with Triaxial Accelerometry”. This is an interesting research about the performance of footwork of flamenco dance (Zapateado – 3) and the influence in the human musculoskeletal.

The manuscript is well written, structured but changes are required.

0.      Abstract: The abstract meets requirements

1.      Introduction:

The introduction references are not published in scientific journals. Please actualize this. Ref: 1,2,4,6,7…)

Line 50: Can authors explain better what “playerload” is?

2.      Material and methods:

Can authors explain how calculate the sample size?

The research consisted of two groups, a professional group and an amateur group. What is the difference? Regarding to the age characteristics  professional:38.83±7.96 amateur: 34.50±10.67. The professional are considered with a minimum of 3 years of experience… Is not clear the difference between age and years experience. Please explain better.

Exclusion criteria: Didi authors take care about foot morphology? Like hallux abductus valgus, cavus feet, planus feet….

Table 1: Please add abbreviations and the meaning of m Kg…

Line 106: Explain better akimbo posture

Table 2: please add abbreviations Hz and BPM

Statistical analysis: please add references.

3.      Results:

Table 3: Can authors explain better the results of the table 3 in text?. Where are the p values? The results are not concordance with the table 3 data.

4.      Discussion:

Lines 274-277: How can influence the foot morphology in the research? Did authors take care about this?

Line 305: Where participants performance the Zap-3? Who was the surface?

Please, add Limitations of the research.

5.      Conclusion.

Authors should rewrite the conclusion pf the research. The conclusion should answer the research question. Please erase  “future studies” this should be in discussion.

Author Response

Thank you for the opportunity to amend the manuscript.

Thanks a lot for your time.

Round 2

Reviewer 1 Report

The authors improved the manuscript and it can be accepted.